# Large-Scale Climatic Patterns Have Stronger Carry-Over Effects than Local Temperatures on Spring Phenology of Long-Distance Passerine Migrants between Europe and Africa

**DOI:** 10.3390/ani12131732

**Published:** 2022-07-05

**Authors:** Magdalena Remisiewicz, Les G. Underhill

**Affiliations:** 1Bird Migration Research Station, Faculty of Biology, University of Gdańsk, Wita Stwosza 59, 80-308 Gdańsk, Poland; 2Department of Biological Sciences, University of Cape Town, Rondebosch, Cape Town 7701, South Africa; les.underhill@uct.ac.za; 3Biodiversity and Development Institute, 25 Old Farm Road, Rondebosch, Cape Town 7700, South Africa

**Keywords:** climate change, passerine migration, spring phenology, long-distance migrants, large-scale climate indices, NAO, IOD, SOI, Europe, Africa

## Abstract

**Simple Summary:**

Spring in Europe has been trending earlier for almost half a century. Long-distance migrant birds, such as the Willow Warbler and Pied Flycatcher, which breed in Europe, have arrived earlier too. It is broadly accepted that warming springs in temperate regions explain the earlier arrival of migrants. However, migration started weeks earlier and thousands of kilometres away. There must be additional cues elsewhere triggering migration. Meteorologists have developed measures of atmospheric circulation which are related to climate variability in wide regions. One of them is the Southern Oscillation Index, which reflects El Niño/La Niña that cause droughts and floods in the southern hemisphere. Other atmospheric circulation patterns, measured by the North Atlantic Oscillation Index and Indian Ocean Dipole, help predict total rainfall for a whole season in various parts of Africa and Europe. Good rains are associated with plant growth and with insect abundance. Insects provide food for most of these migrants. Therefore, this paper asks the question: “Is the timing of arrival of long-distance migrants in spring related to the climates they experience in the places where they are over the year prior to arrival in Europe?” This paper says the answer is “Yes”.

**Abstract:**

Earlier springs in temperate regions since the 1980s, attributed to climate change, are thought to influence the earlier arrival of long-distance migrant passerines. However, this migration was initiated weeks earlier in Africa, where the Southern Oscillation, Indian Ocean Dipole, North Atlantic Oscillation drive climatic variability, and may additionally influence the migrants. Multiple regressions investigated whether 15 indices of climate in Africa and Europe explained the variability in timing of arrival for seven trans-Saharan migrants. Our response variable was Annual Anomaly (AA), derived from standardized mistnetting from 1982–2021 at Bukowo, Polish Baltic Sea. For each species, the best models explained a considerable part of the annual variation in the timing of spring’s arrival by two to seven climate variables. For five species, the models included variables related to temperature or precipitation in the Sahel. Similarly, the models included variables related to the North Atlantic Oscillation (for four species), Indian Ocean Dipole (three), and Southern Oscillation (three). All included the Scandinavian Pattern in the previous summer. Our conclusion is that climate variables operating on long-distance migrants in the areas where they are present in the preceding year drive the phenological variation of spring migration. These results have implications for our understanding of carry-over effects.

## 1. Introduction

Many migrant birds have been arriving earlier in spring in Europe and in North America since about the 1980s; this has mostly been attributed to increased spring temperatures at their northern stopovers and breeding grounds—an influence of climate change [1,2,3,4,5,6,7,8,9,10,11,12,13]. However, long-distance migrants to Europe begin their spring journey to the breeding grounds several weeks earlier, at wintering grounds in Africa; thus, climatic variation at these remote non-breeding grounds also influences the spring timing of these species [2,13,14,15,16,17,18,19,20,21,22,23,24,25,26,27].

Long-distance migrants often use extensive wintering grounds that may span the Mediterranean coasts of Europe and Africa and sub-Saharan Africa from the west to the east, e.g., Willow Warbler *Phylloscopus trochilus*, Chiffchaff *P. collybita*, Common Redstart *Phoenicurus phoenicurus*, Blackcap *Sylvia atricapilla*, Garden Warbler *S. borin*, Barn Swallow *Hirundo rustica*, European Reed Warbler *Acrocephalus scirpaceus*, Sedge Warbler *A. schoenobaenus*, Red-backed Shrike *Lanius collurio*, Spotted Flycatcher *Muscicapa striata* [28,29,30,31,32]. However, most studies that have related migrants’ spring arrivals to conditions at the non-breeding grounds have focused on climatic variation in western Africa and southwestern Europe, likely because they have mostly analysed data from western Europe [2,15,16,17,18,24,25,33,34,35,36,37,38,39,40,41,42]. This is understandable, given that most migrants breeding in south-western Europe winter in western Africa [29,30,31,32]. Fewer studies have related the timing of spring’s arrival in Europe of long-distance migrants to climate and habitat variability in eastern and south-eastern Africa [2,19,20,21,26,43]. Even fewer studies have shown that climatic variation in different parts of Africa [2,22,23,26], or in both hemispheres [2,24,44,45,46], might jointly influence spring arrivals of migrants to Europe for breeding.

Many studies have demonstrated relationships between migrants’ spring arrivals in Europe and temperatures at their non-breeding areas [2,14,16,19,24,26,27,40,47,48,49,50]. The precipitation which migrants experience at their wintering grounds and stopover areas has been shown to influence their spring arrival in Europe [2,14,16,19,24,25,26,27,40,48,49,51,52]. Climate variability in Africa and Europe, including rainfall and temperatures, is largely shaped by ocean–atmospheric interactions; thus, large-scale climate indices that reflect these interactions, such as the Northern Atlantic Oscillation Index (NAOI), the Southern Oscillation Index (SOI/ENSO), and the Indian Ocean Dipole (IOD), might be used as convenient proxies for ecological conditions that the migrant birds experience over wide non-breeding grounds at different stages of their life [2,15,21,22,23,26,35,36,37,43,53]. The Northern Atlantic Oscillation (NAO) and the Scandinavian Pattern (SCAND) are proxies for conditions on the breeding grounds in north-western, central, and north-eastern Europe for many long-distance migrants [22,23,45,54].

We have shown that, in one long-distance migrant, the Willow Warbler, the combined effect of multiple indices of climate at both its breeding grounds in Europe and its non-breeding grounds in western, eastern, and southern Africa explained not only the long-term trend to earlier arrival, but also nearly 60% of the year-to-year variation in timing of this species’ spring migration in 1982–2017 at the Polish coast of the Baltic Sea [22,23]. Thus, we also expect that, in other long-distance migrants, a combination of several large-scale and local climate indices should help explain their spring arrival timing in Europe. In this overview we aim to identify the combination of large-scale indices of climate in Africa and in Europe which explain long-term trends and year-to-year variation in the timing of spring arrivals at the Baltic coast over 40 years (1982–2021) for seven long-distance migrant passerines. We chose species on spring passage through the Baltic coast between their wide wintering grounds in western, eastern, and south-eastern Africa and their breeding grounds in northern Europe. We also provide an overview of these climate indices and discuss the ways in which they might influence the timing of spring migrants. Understanding the combined influences of climate variability at both hemispheres is crucial for understanding the drivers for the phenological shifts caused by the climate change in long-distance migrants within the Palaearctic-African Bird Migration System.

## 2. Materials and Methods

### 2.1. Species Selected for Analyses

To evaluate the common patterns of the influence of the large-scale climate indices on migrant birds, we selected all species of long-distance migrant passerines which were caught in spring in relatively large samples sufficient for analysis at the Bukowo ringing station over 1982–2021. These seven species are: Blackcap *Sylvia atricapilla*, Lesser Whitethroat *Curruca curruca*, Common Whitethroat *Curruca communis*, Willow Warbler *Phylloscopus trochilus*, Redstart *Phoenicurus phoenicurus*, and Pied Flycatcher *Ficedula hypoleuca*, Chiffchaff *Phylloscopus collybita*. These species have wide breeding grounds in Europe and are trans-Saharan migrants (Figure 1). Their extensive non-breeding grounds span western and central Africa, and in six species also eastern Africa; the Willow Warbler’s non-breeding grounds extend to southern Africa. The routes of populations that migrate in spring from different regions in Africa to their breeding grounds in Scandinavia cross at Bukowo [37]. Migration distance in these species between their farthest wintering grounds in Africa and breeding grounds in Scandinavia and northern Russia varies from about 5000 km for the Lesser Whitethroat to nearly 12,000 km for the Willow Warbler (Figure 1) [25,34,37,39,42,55,56,57,58,59,60].

### 2.2. Study Site and Sampling

We used daily numbers of the seven long-distance migrants mistnetted and ringed during spring migration over four decades (1982–2021) at the Bukowo ringing station on the Baltic Sea coast (54°20′13″–54°27′11″ N, 16°14′36″–16°24′08″ E) (Appendix A, Figure A1 and Figure A2). During each spring, ringing was conducted daily from 23 March–15 May; this period covered the spring migration of the species in focus. The number of 8 m-long nets was stable within each spring, though ranged from 35 to 57 in different years. Mistnetting and ringing followed the standardised monitoring protocols of the Operation Baltic project [61,62]. More detail on methods of bird ringing at Bukowo is provided in the Appendix A. In spring, all Willow Warblers were in the same plumage and thus were aged as “full-grown” [63,64]. In the remaining six study species, most could be aged as “young“ (birds in their first year of life) or “adult” (older), but some individuals could only be aged as “full-grown” [63,64]. For consistency of analysis, we treated all age categories jointly for each of the seven species, considering that most individuals, including young birds, were likely to attempt breeding during the spring of capture [56]. In the analyses we considered only the first capture of an individual in each season. Catching and ringing of birds was conducted with the annual approval of the Polish Academy of Sciences and the approval of the General Directorate for Environmental Protection, Poland (last decision: DZP-WG.6401.102.2020.TŁ). Field research at Bukowo was approved annually by the Marine Office in Słupsk or Szczecin (last decision: OW.5101.42.21.DSz(9)).

### 2.3. Calculating the Annual Anomaly of Spring Migration at Bukowo for Each Species

We analysed the numbers of birds of the seven species caught each day during 23 March–15 May at Bukowo in 1982–2021 (Appendix B, Table A1). We excluded years in which fewer than 10 individuals of a species were caught; thus, in six species, we analysed data for 32–38 years during 1982–2021; for the Common Whitethroat, 19 years had sufficient data (Appendix B, Table A1). For each species, the daily totals of birds caught were recalculated to daily percentages of the total number of individuals caught that spring. These daily percentages in subsequent days of spring were summed to a cumulative arrival curve for the season for each species. The values from these cumulative curves for each day were averaged across the years analysed to produce the long-term arrival curve for each species (Figure 2) in which each year had the same weight.

Next, for each species, we calculated the annual anomaly (AA) for each year as the departure of the cumulative curve for this spring from the average cumulative curve of spring passage in 1982–2021 [22]. Negative and positive values of AA indicated, respectively, earlier and later passage of spring than the long-term average curve for the species in focus. We then used the annual anomalies (AA) for each species’ spring migration as the response variable in multiple regression models, with 15 climate indices and calendar year as explanatory variables; we used the same variables as in [23]. These were based on groups of months pertinent to the birds’ life-stages during the year preceding the spring migration (Figure 3); see [23] and Section 2.4 and Section 2.5 for details.

### 2.4. Ranges of Months When Each Climate Index Might Influence a Migrant

The seven long-distance migrants spend only three to four months at the breeding grounds; migration takes four to five months of the year, and they spend the remaining four months at their non-breeding grounds, where they undergo a complete moult [56]. The timing of these life stages can differ a few weeks for different populations of a species, and some individuals might still be on migration when the others already breed, and some birds might have left the wintering grounds, but the others are on transit through the wintering grounds [56]. We aimed to analyse the influence of the climate indices on the migrants in the whole period when these species are observed at the non-breeding grounds (November–March) and to focus on the main part of the breeding season (June–July). Thus, we divided the year into such periods of the year (Figure 3) to enable analysis of any common relationships between timing of spring migration in these species and the climate indices that influence the birds in different parts of their ranges (Figure 1) where they stay at subsequent life stages.

Thus, we considered April–May to be “spring migration”, June–July to be “the main part of the breeding season”, August–October to be “autumn migration”, and “non-breeding season” to be November–March (Figure 3), using northern hemisphere seasons (Figure 3). This division matches the timing of the life stages of our seven species to within half a month [56]. We analysed the effects of the climate indices only in these periods of the year when the migrants occur in the regions where the climate indices operate (Table 1), as in our earlier papers [22,23].

### 2.5. Climate Indices

Meteorologists have shown that the North Atlantic Oscillation, the Southern Oscillation, and the Indian Ocean Dipole are the meaningful proxy ocean–atmosphere features which influence climate, including rainfall, temperature, and wind patterns, in Europe and Africa [53,65,66,67,68]. The Scandinavian Pattern contributes to climate variability in northern Europe [69]. We used a combination of these large-scale climatic indices (Table 1, Figure 4) and local temperature and precipitation values for areas where large-scale indices were not available to investigate the influence of climatic variability on the spring passage of seven long-distance migrants.

#### 2.5.1. North Atlantic Oscillation (NAO)

The North Atlantic Oscillation (NAO) influences year-to-year variability in the climate of the Northern Hemisphere [43,66]. The NAO is formed by the differences in the positions of the Icelandic Low and the Azores High; these shape the strength of westerly winds and storm tracks over the North Atlantic, and the associated patterns of temperature and precipitation in western and central Europe and northwestern Africa (Figure 4A,B) [43,66,68,70]. The NAO Index (NAOI) is calculated as the differences in the normalized sea level pressures (SLP) between the weather stations in the Azores (Ponta Delgada) in the central North Atlantic and in south-western Iceland (Reykjavik) [66,71,72]. The positive phase of NAO (NAO+) represents a stronger than usual difference in pressure between the two regions and is associated with strong westerly winds that bring wet, mild, and stormy weather to Europe. Thus, NAO+ in November–March (winter NAO) is related with wet and warm winters and early springs in northern Europe and dry spells in the Mediterranean region, including north Africa (Figure 4A) [43,66,72]. The negative phase of winter NAO is associated with the opposite conditions. The summer NAO is the dominant factor of variability of summer climate in the Atlantic region of Europe (Figure 4B) [73,74]. A positive NAO increases the risk of extremely dry and hot weather over the European coasts, as in summer 2018, and a negative summer NAO may cause such conditions in the Mediterranean region, as in 2019 [74]. We used the monthly NAO Index (NAOI), normalized using the monthly means and standard deviations, for the 1981–2010 baseline period (Table 1), as provided by the US National Oceanic and Atmospheric Administration, National Weather Service, Climate Prediction Center [70]. NAOI for November–March provides a good indication of weather during winter and early spring over northwest Europe, but NAOI for April–September should be interpreted more cautiously [66,68,72,75]. NAOI in June–July showed a nearly significant decreasing trend over 1982–2021, but no long-term trends in other months (Figure 5; Appendix B, Table A2).

#### 2.5.2. Indian Ocean Dipole (IOD)

The Indian Ocean Dipole, described as recently as 1999, is an irregular oscillation of the Sea Surface Temperature (SST) in the western and the eastern Indian Ocean [53,67]. The measure for the IOD (Table 1) is the Dipole Mode Index (DMI), which is the gradient between the anomalies of SST between the western (50°–0° E) and the eastern (90°–110° E) regions of the Indian Ocean around the equator (10°S–10° N). The positive values of DMI indicate the positive phase of the Indian Ocean Dipole (IOD+), when the western Indian Ocean is warmer than its eastern region. This causes easterly winds that bring above-average rainfall during the East African Short Rains in October–December (Figure 4A) [53,67]. Torrential rains in some years cause floods in East Africa, such as those in the end of 2019 [76]. The negative phase of the IOD (IOD−) reflects the opposite gradient of temperatures in the Indian Ocean, which is related with westerly winds that drive moisture away of East Africa [53,67,70,77,78]. The Indian Ocean Dipole usually varies independently of the Southern Oscillation [79,80], but the interaction between IOD and ENSO can generate Super El Niños, which occurred in 1972/1973, 1982/1983, and 1997/1998 [81]. The IOD in November–March and in August–October showed a distinct increasing trend over 1982–2021 (Figure 5; Appendix B, Table A2).

#### 2.5.3. Southern Oscillation (SOI)

The Southern Oscillation (SO) is the large-scale atmospheric fluctuation in air pressure between the western and eastern tropical Pacific during El Niño (EN) and La Niña episodes. The El Niño/Southern Oscillation (ENSO) and La Niña are cyclical oceanographic phenomena, caused by movements of warm sea surface temperatures (SST) across the Pacific Ocean; they strongly influence the climate in eastern and southern Africa (Figure 4A) [53,82,83,84]. The measure of this pattern is the Southern Oscillation Index (SOI), which is a standardized index based on the differences in the air pressure at the sea level between Tahiti at the west and Darwin, Australia, at the east of the Pacific. The positive (cold) phase of the SOI indicates La Niña episodes, with the above-average air pressure at the west and the below-average pressure at the east. During La Niña, the central Indian Ocean is colder than usual, which results in strong easterly winds that bring moisture into Africa [53,70]. La Niña is thus associated with wet and cool weather in eastern and south-eastern Africa [85,86]. The negative (warm) phase of the SOI indicates El Niño episodes, with warm waters in the central Indian Ocean. This pattern results in below-average differences in temperatures between the ocean and land of East Africa, which causes weaker than average equatorial easterly winds [53,70]. Such atmospheric pattern brings warmer and drier than average weather in central Tanzania, but warmer and wetter conditions in the region of the Lake Victoria (Figure 4A), especially during the October–December rainy season [53]. El Niño (SOI−) is related with hot and dry conditions in south-eastern Africa, such as those that caused the 2015/2016 drought [87]; La Niña (SOI+) brings moisture and cold to this region, such as those at the turn of 2021/2022 [70]. SOI in November–March had a near-significant increasing trend over our study period, but no trend in August–October (Figure 5; Appendix B, Table A2).

#### 2.5.4. Sahel Precipitation Index (PSAH)

Rainfall and temperature in the Sahel region are influenced by several ocean–atmosphere oscillations, such as NAO and the Atlantic Multidecadal Oscillation (AMO) [88,89] and in the eastern part, probably by IOD [53], and thus no single convenient large-scale climate index was available to reflect conditions there. Therefore, we used the Sahel precipitation anomaly (PSAH) within 10–20° N and 20° W–10° E, based on the reanalysis of data from the weather stations in this region (Table 1, Figure 4A) as the index of rainfall in the western Sahel, where many long-distance migrants, including our seven study species, overwinter (Figure 1). Positive monthly values of PSAH indicate higher rainfall than the 1982–2021 average in that month, and negative values indicate below-average rainfall [90]. We chose these boundaries for consistency with the area covered by the Sahel Precipitation Index (SPI) provided for 1950–2017 by the Joint Institute for the Study of the Atmosphere and Ocean (JISAO) [91]. The values of PSAH from Climate Explorer by the World Meteorological Organisation, which we used here (Table 1), were almost perfectly correlated (r = 0.98, *p* < 0.0001) with the SPI in 1982–2017 from JISAO, which justified comparing our current results with earlier studies that used SPI [22,25,26,37,92]. The main period of precipitation in the Sahel is June–October [91], and summer rainfall in this region has increased since the 1980s, caused by enhancements in the African easterly jet; this improved moisture has been related to an increase in vegetation and “greening” of the Sahel [93]. This increase is demonstrated by the significant trend in PSAH in August–October over 1982–2021, but there was no trend in November–March, which is the dry season in the Sahel (Figure 5; Appendix B, Table A2).

#### 2.5.5. Sahel Temperature Anomaly (TSAH)

As an indication of the temperatures in the western Sahel, we used the Sahel temperature anomaly (TSAH), based on the air temperature at 2 m above the ground level from the ERA5 dataset, which is a reanalysis of climate data that combines observations from weather stations with estimates from weather models to fill in gaps, to provide values at 31 km grid resolution for each hour (Table 1). We used the same range of coordinates as for the Sahel Precipitation Index (PSAH) (Figure 4A). The positive monthly values of TSAH indicate higher temperatures than the 1982–2021 average, and *vice versa*. The Sahel temperatures in August–October and November–March increased by 0.6 °C and 1.3 °C, respectively, over the 40 years of our study (Figure 5; Appendix B, Table A2).

#### 2.5.6. Local Temperatures at the Baltic Coast (TLEB)

We used temperatures during spring (April–May) from Łeba (54°45′ N, 17°32′ E) (Figure 4B), the closest coastal weather station to Bukowo, as the proxy for the local conditions that the birds encounter when they arrive at the Baltic coast, where we monitored them (Figure 1, Appendix A Figure A1). We calculated the mean bi-monthly temperature for April–May from the daily mean temperatures observed in this weather station provided by the European Climate Assessment and Dataset (http://www.ecad.eu, accessed on 30 June 2022) [94]. Spring temperatures in Łeba increased on average by 1.5 °C over 1982–2017 [23]; however, the trend was not significant over 1982–2021, likely because of cold springs 2019–2021 (Figure 5; Appendix B, Table A2).

#### 2.5.7. Scandinavian Pattern (SCAND)

The Scandinavian Pattern reflects the primary atmospheric circulation over Scandinavia [54,70] with the centres of the opposite pressure systems over northern Europe and central Asia (Figure 4B). The positive phase of this pattern (SCAND+) is associated with the centre of the height over Scandinavia. In spring and summer, such pattern is related to warm and dry conditions over Scandinavia, but below-average temperatures and above-average precipitation over western and central Europe and the British Isles [54]. SCAND showed no trends over 1982–2021 (Figure 5; Appendix B, Table A2).

### 2.6. Statistical Analyses

We selected 15 climate variables and calendar year of spring migration (Table 1) as explanatory variables in multiple regression models to determine if they had any combined effects on the timing of spring migration across the southern Baltic Sea coast, as reflected by the Annual Anomalies for the seven species of long-distance migrants. We included year as an explanatory variable because, if it was included in the final model in the variable selection process, it would imply that the climate variables alone were inadequate to explain the trend in the annual anomalies. In this way, we tested any possible long-term trends and the year-to-year variation in one analysis. To avoid harmful effects of multicollinearity on the results of our models, we checked Variance Inflation Factors (VIF) in our multiple regression models [95]. Details of the use of multiple regressions are provided in our earlier studies [22,23]. We standardized response and explanatory variables so that they had a mean of 0 and a standard deviation of 1 to facilitate interpretation of the results. We used “all subsets regression” by fitting with all the possible combinations of one to 16 explanatory variables, and we selected the best-fitting model, using model ranking by Akaike Information Criteria corrected for small sample size (*AICc*), with the package “MuMIn 1.43.6” [96]. For each climate index selected in the best model for each species, we calculated the partial correlation coefficient (*pR*), which reflects the correlation between the response variable (AA) and each explanatory variable (climate index) while removing the effects of the remaining variables using the package “ppcor 1.1” [97]. To visualise the influence of the climate variables on the timing of spring migration for each species, we present bar graphs which display these partial correlation coefficients. The statistical analyses were conducted in R 4.0.3 [98].

## 3. Results and Discussion

### 3.1. Long-Term Trends in Timing of Migrants’ Spring Passage at the Baltic Sea Coast

Spring migration at Bukowo shifted significantly earlier over 1982–2021 in two of the seven species we analysed, in line with the general trend in migrant passerines for earlier spring arrivals in Europe since the 1980s [1,2,3,6,8,11,99]. For the Blackcap, spring passage shifted earlier by 6.0 days on average, and, for the Willow Warbler, by 5.2 days, during the 40 years of our study, as measured by the Annual Anomaly (AA) (Figure 6; Appendix B, Table A3).

This corresponds with trends for earlier arrivals in both species in other sites in the Baltic region, such as Rybachy, Kaliningrad region of Russia, in 1959–1990 [1], and the island of Christiansø, Denmark, in 1976–1997 [8], as well as at Helgoland, North Sea, Germany, in 1960–2002 [6]. Willow Warblers also advanced their spring arrivals in Gotland, Sweden, in 1990–2012 [100], in 1971–2000 in Oxfordshire, UK [2], and in 1955–2014 at Fair Isle, Scotland [11]. For the five remaining species, the long-term trends at Bukowo were not statistically significant but showed a tendency for earlier arrivals (Figure 6; Appendix B, Table A3), consistent with trends of these species in the other bird observatories [1,6,8]. The less pronounced trends in our study might result from different time periods covered by these studies and our analyses, which included more recent years. The linear trends, significant in three species, explained a maximum of 21% of the variation in their AAs (Figure 6; Appendix B, Table A3). Thus, we set out to determine whether we could explain a substantial percentage of the annual variation in migrant’s spring phenology by selecting a set of climate variables, which might influence the birds at different stages of their annual cycle.

### 3.2. Relationships between Timing of Migrants’ Spring Arrival in Europe and the Large-Scale Climate Indices

#### 3.2.1. Combined Influence of Climate Indices on Spring Migration of the Seven Migrants at Bukowo

All correlation coefficients between the 15 climate indices which we used were |r| < 0.7 (Appendix B, Table A4). The full multiple regression model with AA as the response variable had 16 explanatory variables (15 climate variables and year) (Appendix B, Table A5, Table A6, Table A7, Table A8, Table A9, Table A10 and Table A11). The best models with the smallest AICc value in the “all-subsets” approach explained 27.3–62.4% of the variation in the Annual Anomaly of spring migration for the seven migrant species by a combination of two to seven climate indices (Figure 7; Appendix B, Table A12, Table A13, Table A14, Table A15, Table A16, Table A17, Table A18, Table A19, Table A20, Table A21, Table A22, Table A23, Table A24 and Table A25). All Variance Inflation Factors in our best multiple regression models were VIF < 10 (Appendix B, Table A12, Table A13, Table A14, Table A15, Table A16, Table A17 and Table A18), which indicated no harmful effects of multicollinearity on our results [95]. The best models for the seven species met the assumptions of multiple linear regression [101], as indicated by an inspection of the residuals (Appendix B, Figure 3, Figure 4, Figure A5, Figure A6, Figure A7, Figure A8 and Figure A9), and thus they were satisfactory in explaining the variation in AA. In the figures showing model diagnostics, the residuals vs. fitted values’ plots (top left panels at Figure A3, Figure A4, Figure A5, Figure A6, Figure A7, Figure A8 and Figure A9) for all seven species showed scattered values with no pattern. Likewise, the plots of square root of residuals vs. fitted values (Appendix B, Figure A3, Figure A4, Figure A5, Figure A6, Figure A7, Figure A8 and Figure A9, bottom left panels) showed no consistent upward trend; together, these observations indicated that the variance did not increase with the mean [101]. The pattern in the “normal Q-Q” plots (Appendix B, Figure A3, Figure A4, Figure A5, Figure A6, Figure A7, Figure A8 and Figure A9, top right panels) indicated that the standard errors had, at least approximately, normal distributions [101]. In the plots of residuals vs. leverage (Appendix B, Figure A3, Figure A4, Figure A5, Figure A6, Figure A7, Figure A8 and Figure A9, bottom right panels), none of the points lay beyond the contour for the Cook’s distance = 1, which indicated that no single observation had any critical influence on model estimates [101] (Appendix B, Figure A3, Figure A4, Figure A5, Figure A6, Figure A7, Figure A8 and Figure A9). The year, as an explanatory variable, when used in combination with the other climate variables, was not selected in any of the best models (Figure 7; Appendix B, Table A12, Table A13, Table A14, Table A15, Table A16, Table A17 and Table A18), which indicated that long-term temporal effects were absorbed by the climate variables, so that these accounted for the trends in the timing of migration, especially in the two species with significant trends (Figure 6). The signs of most of the partial correlation coefficients (*pR*) were negative (Figure 7; Appendix B, Table A12, Table A13, Table A14, Table A15, Table A16, Table A17 and Table A18), which implies an earlier passage with the larger value of a climate index.

#### 3.2.2. Influence of the North Atlantic Oscillation (NAO)

Numerous studies showed that many species of passerines arrive in northern Europe early after a positive NAO in the previous winter (December–March), which is related to mild winters and early and warm springs in Europe [1,2,4,6,21,27,35,36,37,43,96,99,100,101,102,103,104,105]. In line with these results, spring passage at Bukowo was early when the NAO for November–March or August–October of the previous year was large, and vice versa in four of the seven species we analysed (Figure 7). Common Redstarts, Pied Flycatchers, and Willow Warblers arrived at Bukowo early with a largely positive NAO in November–March, and Common Whitethroats after a largely positive NAO in August–October of the previous year (Figure 7). Blackcaps, Lesser Whitethroats, Common Whitethroats, and Willow Warblers migrated through Bukowo early with a largely positive NAO in April–May (Figure 7), a pattern related with warm springs in Europe (Figure 4B). The Lesser Whitethroat and Willow Warbler spring arrivals at Bukowo were also early for the NAO in June–July, though the previous year was small (Figure 7). This pattern is related to dry summers in north-western Europe [66], which should provide favourable conditions for raising offspring by passerines [106]. However, the winter NAO index explained only 0–6% of the variance in the phenology of spring migration in 23 species at Helgoland, Germany; in 46% of the papers reviewed in this study, winter NAO showed no significant relationship with the phenology of birds’ spring arrivals in Europe [107]. Those results, as well as our study, show that NAO does influence the passage of some species, but has no influence on the others, probably depending on location of wintering grounds and stopovers of different populations in relation to the area under climatic influence of this index (Figure 4A,B). Our results showed that other climate patterns, such as IOD, SOI, or SCAND, often have a greater influence on the spring phenology of migrants arriving in Europe than NAO (Figure 7, Table A12, Table A13, Table A14, Table A15, Table A16, Table A17 and Table A18). We thus suggest that the effects of NAO on birds should be analysed in combination with other large-scale climate indices.

#### 3.2.3. Influence of the Indian Ocean Dipole (IOD)

IOD has been identified more recently than the other oscillations [53]; thus, there are limited studies of its relation to the ecology of migrants [20,21,22,23,26,108,109]. White Storks *Ciconia ciconia* breeding in Poland had smaller clutch sizes and smaller eggs after low IOD in August; this relates to low rainfall at the eastern African non-breeding grounds of the species, and is probably a carry-over effect on the female condition [26]. Similarly, IOD explained a part of variation in the dates of the first clutches of Red-backed Shrikes in Czech Republic, probably because this climate index is related with precipitation, and thus insect availability, at the species’ non-breeding grounds in south-eastern Africa [20]. Conditions at the stopover sites of migrants in eastern Africa, where IOD operates (Figure 4A), influence the timing of migrants’ spring arrival in Europe, as demonstrated by late arrivals in spring 2011 in Denmark of the Red-backed Shrike and the Thrush Nightingale *Luscinia megarhynchos* after drought in the Horn of Africa [19], reflected by a low IOD value for 2011 (Figure 5). Analogously, in five out of the seven long-distance migrants we analysed, spring passage was related with IOD in August–October and/or November–March, as shown by pink bars in Figure 7. Common Whitethroat, Chiffchaff, Willow Warbler, and Pied Flycatcher migrated through Bukowo late with low IOD value in August–October (Figure 7, Table A14, Table A15, Table A16, Table A17 and Table A18), which indicated low rainfall in East Africa. However, the Common Redstart and Pied Flycatcher also arrived at Bukowo late with high IOD in November–December, and thus high rainfall in the late part of the short rains period in East Africa (Figure 7; Appendix B, Table A15 and Table A18). We suggest that this pattern, seemingly in contradiction with other migrants’ late arrival after drought, might be explained by the locust plague in East Africa and the Middle East and, such as that in 2019/2020, can be attributed to heavy rains in the end of the year [110]. Locusts, too large a prey for these small birds, would, however, consume their food at their non-breeding sites, which might result in the poor condition of migrants departing from these areas, or force them to overwinter farther south than in other years, hence their late arrival at the Baltic coast. The IOD probably also affects movements of seabirds, because the production of plankton in the western Indian Ocean—food for some species—tends to be smaller with a larger DMI value [109]. Food shortage offshore was probably related with the unusually frequent occurrence of the Red-necked Phalaropes *Phalaropus lobatus* at East African estuaries and inland lakes in 2020, following the extreme IOD in 2019/2020 [109]. The increasing trend in the IOD (Figure 5) indicates that SST in the western Indian Ocean has increased; this implies that the likelihood of extreme positive IOD events [111] will increase, with associated consequences for the environment and migrant birds. We suggest that future studies of the effect of climate change on birds in Europe should consider the IOD, because its influence on migrants has been underestimated in the past [21].

#### 3.2.4. Influence of the Southern Oscillation Index (SOI)

In tropical and sub-tropical regions, La Niña events, reflected by large positive SOI (Figure 4A), are associated with widespread rainfall events in eastern and south-eastern Africa and in central and southern America, which result in a rapid increase in primary productivity and insect abundance, and thus favourable conditions for insectivorous birds [112]. The effects of SOI have been investigated mostly in long-distance migrants between the Americas [10,49,113] and in Antarctic birds [114,115], though scarcely for the Afro-Eurasian migrants [22,23,116]. Among the American warblers, the timing of the spring arrival of long-distance migrants at Lake Manitoba, Canada, from wintering grounds south of the US–Mexico border, was earlier, and the birds were in better condition after La Niña than after El Niño events [49]. Similarly, several medium-distance migrants arrived early in spring in Massachusetts, USA, after winters with a positive SOI, related to high rainfall in Central America and the Caribbean, where those species overwinter [10]. Similarly, Willow Warblers, the species with the southernmost wintering range in Africa of the species we analysed, arrived at Bukowo in spring early after large SOI in November–March (Figure 7; Appendix B, Table A17), related with La Niña extensive rainfall in south-eastern Africa. In contrast, the Common Redstart arrived at Bukowo early after low SOI (Figure 7; Appendix B, Table A15), which is related with El Niño and drought in that part of Africa, but also with wet conditions around the Lake Victoria [53] (Figure 4A), in the southernmost part of this species range (Figure 1). However, the influence of SOI on the mean timing of spring passage of 13 trans-Saharan migrant passerines, including our study species, was not significant when it was analysed in combination with temperature or the Normalised Difference Vegetation Index (NDVI), which reflected greenness at areas they visited [116]. This suggests that SOI has an indirect effect on migrants, through its influence on environment conditions directly experienced by the birds. SOI had no influence on the timing of spring arrival of migrants at the region of the Lake Michigan, USA [113], and on the Peregrine Falcon *Falco peregrinus* at the Rankin Inlet, in the Canadian Arctic [117]. However, it seemed to influence the timing of the arrival and reproduction of Antarctic seabirds through its effect on the breakup of the ice, which enables these birds to access the colonies for breeding [115]. We encourage more attention from researchers to the effects of the Southern Oscillation on migration timing and survival of those long-distance migrants that winter in southern Africa.

#### 3.2.5. Influence of Precipitation (PSAH) and Temperatures (TSAH) in the Sahel on Migrants

The Sahel Precipitation Index serves as a proxy for rainfall in the Sahel and the Sahel Temperature Anomaly as a proxy for temperatures in that region, where many European migrants stop-over during migrations or overwinter [22,23,25,26]. Several migrants, including our study species, arrived in spring at Helgoland, Germany, early after wet and warm winters in western Africa [16]. Similarly, for five of the seven focus species, spring migration timing at Bukowo was related with temperature or precipitation in the Sahel in the preceding August–October and November–March periods (Figure 7, green bars). For the Willow Warbler, Common Redstart, and Pied Flycatcher, spring migration at Bukowo was earlier after warmer temperatures in the Sahel in November–March, and for the Blackcap in August–October (Figure 7). This corresponds with the early arrival in spring of the passerine migrants, including our study species, at Capri, Italy, after hot and wet winters in north Africa and the Sahel [27], and in Oxfordshire, UK, after warm temperatures in December–February in Africa south of 20° N [2]. In the Pied Flycatcher, spring arrival was late when in August–October in the western Sahel temperatures were warm (Figure 7). We suggest that warm temperatures without good rainfall in the Sahel might cause dry conditions during stopover and wintering; this would cause poor survival and bad condition.

Good rainfall in the Sahel during winter, reflected by a large PSAH, likely has a positive effect on the condition of departing migrants from this region and thus facilitates their early arrival in spring at the Baltic coast. Lesser Whitethroats, Blackcaps, and Common Redstarts arrived at Bukowo early when there was greater than average rainfall in the western Sahel in August–October and November–March, respectively (Figure 7). This corresponds with the positive relation between Lesser Whitethroat and Common Whitethroat population size and the amount of rainfall in the Sahel [25]. Barn Swallows also arrived in Spain early in the spring after winters with above average rainfall in western Africa [14]. Below average rainfall at autumn stopover sites in Spain and Morocco had negative carry-over effects on European Reed Warblers in terms of reduced survival and late return to the breeding grounds the following spring [118]. Female White Storks in western Poland laid more eggs after an above average Sahel Precipitation Index in August and September, which probably improved food abundance there, and thus in the birds’ condition [26]. In contrast, after drought in the Sahel, mortality of White Storks was larger than after wet years, and was larger for juveniles than for adults, which might have been caused by poor feeding conditions in dry years, or by the enforced southwards shift of the wintering area than in wet years [26]. Moreover, Willow Warblers had a two-fold larger mortality while crossing the Sahara Desert in spring after dry years in the Sahel than in wet years [25].

#### 3.2.6. Influence of the Local Temperatures (TLEB) during Migration

For many passerine migrants in Europe, earlier spring arrivals since the 1980s have been related to an increase in local temperatures at the sites where they were monitored [1,3,6,102,119,120,121]. However, of the seven long-distance migrants we analysed, the selected models showed that local temperatures in spring were related to the timing of migration at Bukowo only in the Lesser Whitethroat; this species arrived early when it was warm in April–May and vice versa (Figure 7, Appendix B, Table A13). In the remaining six species, other climate indices, such as NAO, IOD, and SOI, were selected by the model as the best explanatory variables for the timing of spring arrival (Figure 7; Appendix B, Table A12, Table A13, Table A14, Table A15, Table A16, Table A17 and Table A18). Spring conditions at the Baltic coast, even the most favourable, could not, in themselves, have influenced early arrival at Bukowo because the migrants would have had to initiate migration from their wintering grounds several weeks earlier. However, the temperatures the migrants encountered at the Baltic coast might influence their arrival at the breeding grounds located farther north. Local temperatures have a stronger influence on the timing of spring migration in Europe for short-distance migrants rather than long-distance migrants [34]. Temperatures are correlated over wide areas [122]. Thus, for short-distance migrants, local temperatures in northern Europe are likely to be correlated with temperatures on the wintering areas in the south of the continent. In fact, temperatures at stopover sites in southern Europe have been related to the timing of arrival at more northern sites [14,16,33,123,124]. Taken together, these observations suggest that the response to temperature at earlier stopover sites enables migrants to make final adjustments to the timing of their arrival on the breeding grounds [16,27,34,119]. We thus propose that relationships between the timing of the arrival of migrants and local temperatures at sites where they are monitored may not be cause–effect, but rather indirect effects of the temperatures which the birds encountered at earlier stages of migration.

#### 3.2.7. Influence of the Scandinavian Pattern (SCAND) during Summer and Spring

The positive phase of SCAND brings warm and dry summers to Scandinavia (Figure 4B) [54]; this should provide good feeding conditions for insectivorous birds during their breeding season. Wet and cold weather limits the availability of insects to insectivorous birds and their nestlings, and thus reduces their survival [106]. Five of the species arrived at Bukowo early after high SCAND in June–July of the previous year, and vice versa (Figure 7). Willow Warblers also occurred at Bukowo in greater numbers in springs following summers with a high SCAND than after low SCAND [22]. These findings correspond with the conclusion of Ockendon and co-authors [45] that the conditions on the breeding grounds are the main drivers of changes in the breeding phenology and nesting success for long-distance migrants. The Scandinavian Pattern in April–May was related with the passage of two of our study species, probably through its influence on weather during the birds’ migration (Figure 7) [54]. Blackcaps arrived early at Bukowo when SCAND was above average in spring (Figure 7). This suggests that SCAND+, related with warm and dry conditions in spring over northwestern Europe, facilitates Blackcap arrivals in northern Europe, possibly of the populations that overwinters in southwestern Europe (Figure 1) [37,39]. This result is in concordance with early arrivals of migrants in Europe during warm springs [35,36,121]. However, the arrival of Common Whitethroats at Bukowo was late when SCAND was above average, and vice versa (Figure 7). Common Whitethroats arrive at Bukowo mostly from sub-Saharan western Africa [48]; thus, cold and wet weather in western and central Europe, related to positive SCAND in spring [54], might impede their passage through these areas and delay their occurrence in northern Europe.

### 3.3. Carry-Over Effects of the Condition of Migrants and Timing at Life Stages Prior to Spring Migration Pattern

NAOI, IOD, SOI, PSAH, and TSAH reflect rainfall and temperatures in Africa [43,53,66,90,125]. Rainfall and temperature influence quality of vegetation and the abundance of insects, the main food sources for long-distance migrants, and also influence other ecological conditions which the birds encounter at stopover sites and wintering grounds [25,43,53,77,112,125]. Thus, the conditions which the birds experience at the non-breeding grounds have carry-over effects on subsequent stages of their lives, in this case on the timing of spring arrival and breeding [22,23,49,50,126,127]. In the tropics and sub-tropics, rainfall has a greater effect on food abundance for birds than temperature, in contrast to the temperate zones of the northern hemisphere, where temperatures are more frequently the limiting factor [128,129]. Warm and wet conditions in Africa increase the abundance of insects, many of which swarm after large rainfall events [18,130,131]. Conditions in the non-breeding season influence the migrants directly or indirectly, through a variety of ways:Birds’ condition during stay and on departure from the wintering grounds.Climate conditions at the wintering grounds influence the body condition of birds, which then has carry-over effects on the timing of their departure and their condition on arrival at the breeding grounds [49,50,51,52,126,132]. Abundant food enables migrants to replace their flight feathers efficiently and produce good-quality feathers that will withstand the rigours of migration [48,52,133]. Early moult of flight feathers leaves the migrants sufficient time for efficient pre-migratory fuelling [48,52,133]. During wet winters in the tropics, migrants were able to fuel faster and accumulate larger fuel loads and arrive at the breeding grounds the following spring in better condition than in dry years [24,49,51,134];Timing of departure from the wintering grounds.After wet winters, the migrants are able to depart from the wintering grounds earlier, and thus arrive earlier at the breeding grounds, than after dry winters [22,23,49,51,134]. In both moist and dry wintering habitats, above average rainfall before spring migration advances the migrants’ departure [24,49,50,134,135,136];Duration and frequency of stopovers.Large fuel reserves on departure from the wintering grounds enable birds to migrate faster, with fewer and shorter stopovers [14,25,126]. Favourable conditions during the stopovers on southward migration in northern and eastern Africa, through which insectivorous Lesser Whitethroats and Bonelli’s Warblers *Phylloscopus orientalis* migrate, had a carry-over effect the following spring, more than six months later; early arrival was observed at a stopover site in Eilat, Israel [137]. Unfavourable conditions on route delay passage, as shown by longer stopovers and later spring arrival in the breeding grounds of some migrants after the 2011 drought in the Horn of Africa than in the other years [19]. By adjusting the rate of migration to the environmental conditions encountered at stopover sites, migrants appear to be able to fine-tune their arrival time to the conditions at the breeding grounds [34,116];Survival of migration and overwintering.Below average rainfall in the wintering grounds decreases the survival rates of migrants [25,135,138,139], which would in turn affect the demography of the breeding population [14,116]. After wet years on the wintering grounds with a high survival rate of migrants, individuals of poor quality are able to successfully complete migration, including immatures less experienced than adults, which potentially results in a delay in the overall mean arrival date at northern sites as compared with dry years [22,116]. The decreasing trends of many trans-Saharan migrants have been attributed to droughts in the Sahel [25,140,141].

### 3.4. Combined Effects of Climate Indices on the Patterns of Spring Arrivals of Long-Distance Migrants

The combined effect of variability in a simple set of objectively determined climate factors (Table 1), some with long-term trends and others without, explained between 27% and 62% of the year-to-year variation and the many-year trends in the annual timing of the spring migration of passerines at the Baltic coast (Figure 7, Appendix B, Table A12, Table A13, Table A14, Table A15, Table A16, Table A17 and Table A18). None of the best models included the year, which indicated that the climate indices with long-term trends that were selected by the models explain the temporal trends in the spring arrivals at Bukowo, which we found in three species (Figure 5, Appendix B, Table A3). Such long-term trends in climate indices can induce epigenetic changes in the migratory behaviour, such as the shortening of migration distance or establishing a new wintering quarter as observed in the Blackcap [142,143], which can partly explain the trends we observed and promote even faster shifts towards the early arrivals of migrants. In the Blackcap, the long-term trend for earlier arrivals might be related with an increase in the temperature in the Sahel (TSAH) in August–October over 1982–2021 (Figure 4A, Appendix B, Table A2)—one of the climate factors that influenced spring migration of this species (Figure 7). The trend in Willow Warblers might be associated with an increase in temperatures (TSAH) and precipitation (PSAH) in the Sahel, which likely influence populations wintering in western Africa, and with an increase in the IOD and SOI, which probably influence populations wintering in eastern and south-eastern Africa (Figure 4A; Appendix B, Table A3) [22,23]. Local spring temperatures in Łeba at the Baltic coast showed no trend over 1982–2021 (Figure 5; Appendix B, Table A2), thus the long-term trends for earlier arrival in long-distance migrants cannot be attributed to an increase of local temperatures, as has been suggested in other studies [1,3,6,34,102]. However, temperatures in Łeba had a significant increasing trend over 1982–2017 [22], likely because 2015 and 2016 were among the warmest years in the century globally [144]; however, the trend in local temperatures up to 2021 was not significant due to cold springs in 2020 and 2021 (Figure 5). This influence of temperatures in just two years, which can change the whole multi-year trend, suggests that large year-to-year variation in climate indices might shape the multi-year trends in birds’ phenology differently, depending on the range of analysed years, as in the Song Thrush *Turdus philomelos* at the Baltic coast [4,47]. We suggest that the combined effects of year-to-year variation in various climate indices jointly shape birds’ spring phenology. Linear long-term trends in arrival might reflect the influence of climate indices with long-term trends [22,107]. The year-to-year variation in the timing of birds’ arrivals, on top of any long-term trends, is probably shaped by the combined responses of different cohorts of migrants to the variability of the climate factors they experience in different parts of their geographical range at subsequent life stages prior to spring migration.

## 4. Conclusions

The ocean–atmosphere scientists and meteorologists who developed the large-scale climate indices probably never imagined that their outputs would be inputs into research on bird migration. We have shown, based on evidence from the literature and our results, that the timing of the spring arrival of long-distance migrants in Europe may be shaped by a combination of several climate indices which seem to influence the birds in different parts of their range at previous life-stages. The atmospheric patterns and climatic variability experienced by populations in western, eastern, and southern Africa have combined carry-over effects on the timing of that species’ spring arrivals in Europe. We suggest that, for several species of long-distant migrants, the timing of spring migration is shaped by combinations of climate indices, which explain the year-to-year variation. These climate variables, which display long-term trends, help to explain long-term trends in the spring phenology of migrants. However, there is not a single uniform set of climate indices for all long-distance migrants. Climatic variation at the wintering grounds enables long-distance migrants to depart earlier after favourable conditions. Conditions at stopover sites influence the rate of their spring migration. Warm spring farther north favours the early breeding of migrants. We suggest that climate change at remote non-breeding grounds, in combination with climate change at the northern hemisphere, are relevant drivers of changes in the arrival timing of migrants in Europe.

## Figures and Tables

**Figure 1 animals-12-01732-f001:**
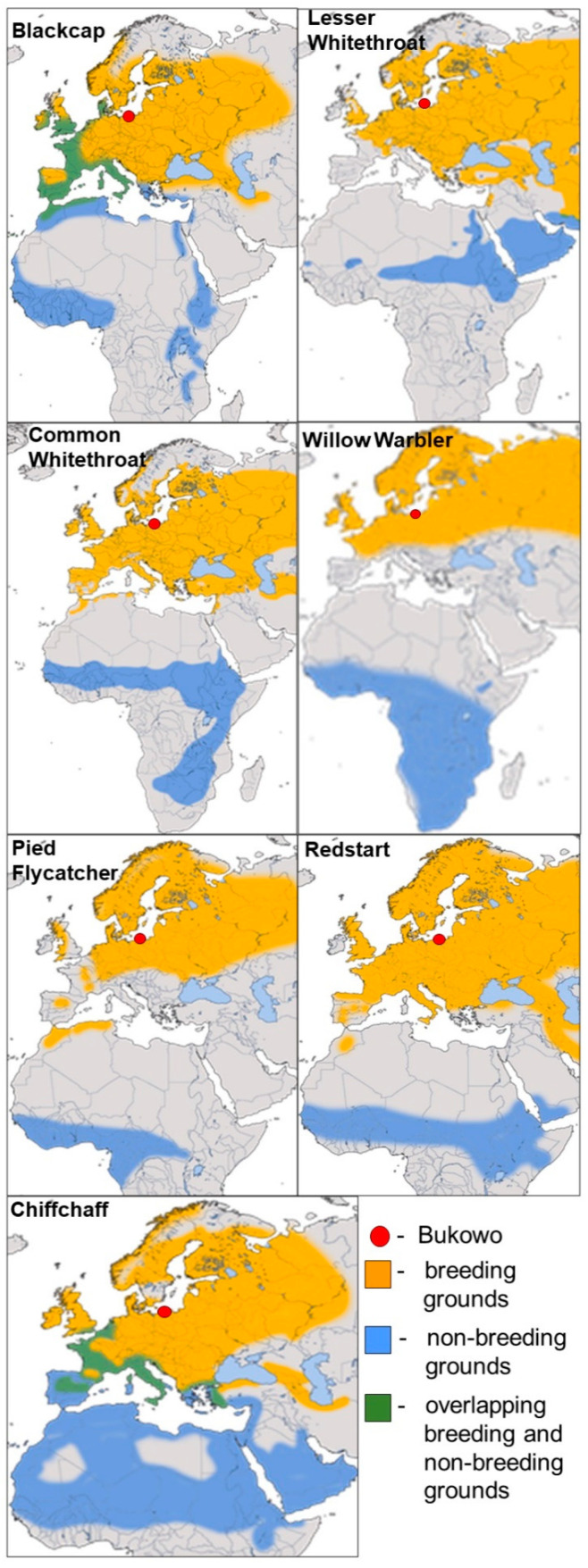
Geographical ranges for the seven long-distance migrants we analysed in the study. Maps by Andreas Trepte after [55], modified.

**Figure 2 animals-12-01732-f002:**
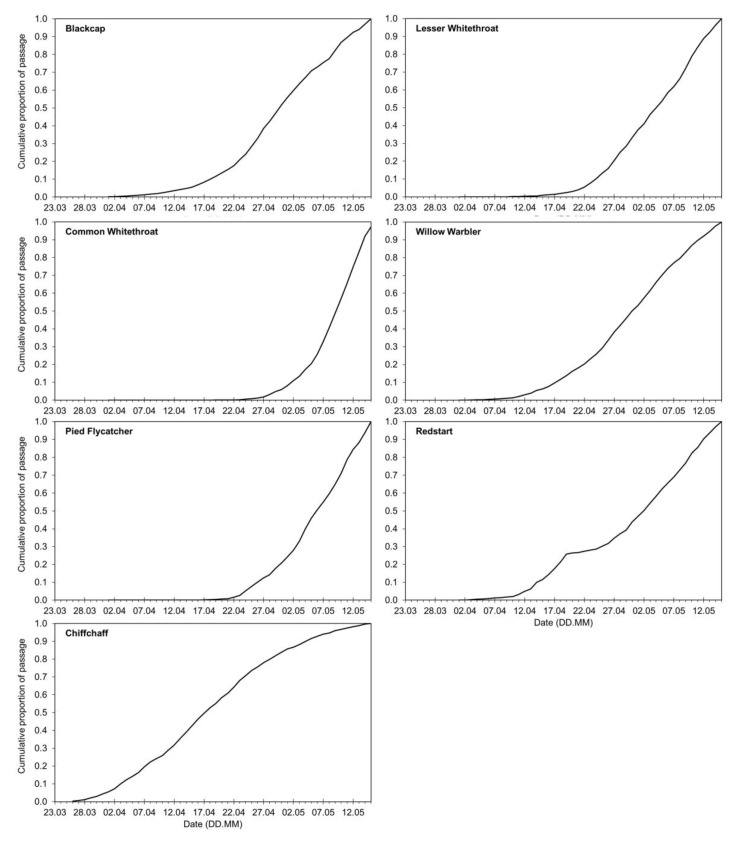
Many-year average cumulative curves of spring arrivals at Bukowo, Poland, over 1982–2021 for the analysed species.

**Figure 3 animals-12-01732-f003:**
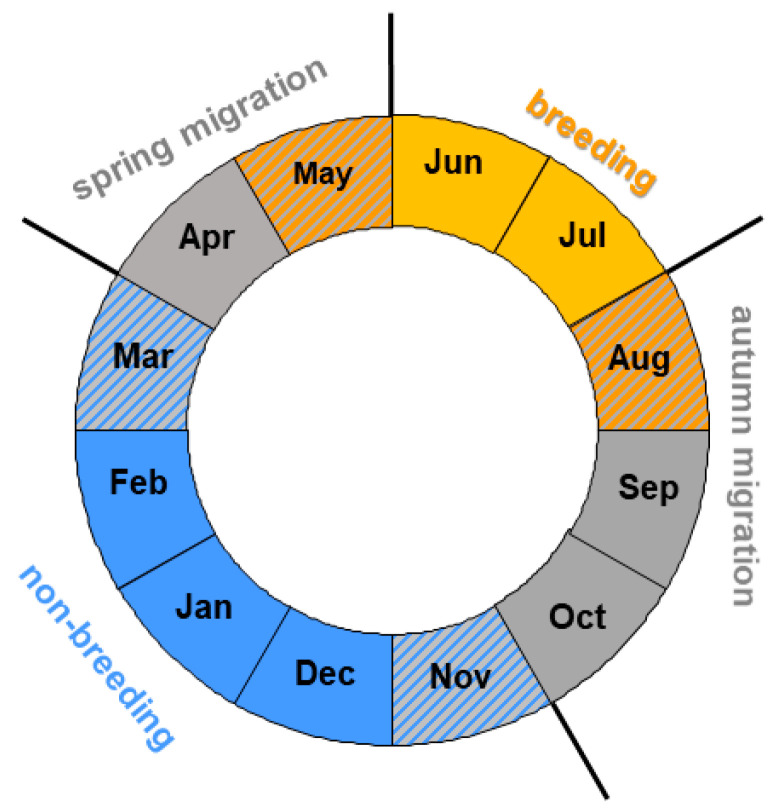
A division of the year into periods applied in the analyses. Fields with two colours indicate possible overlapping of subsequent life stages for different populations of a species of a long-distance migrant.

**Figure 4 animals-12-01732-f004:**
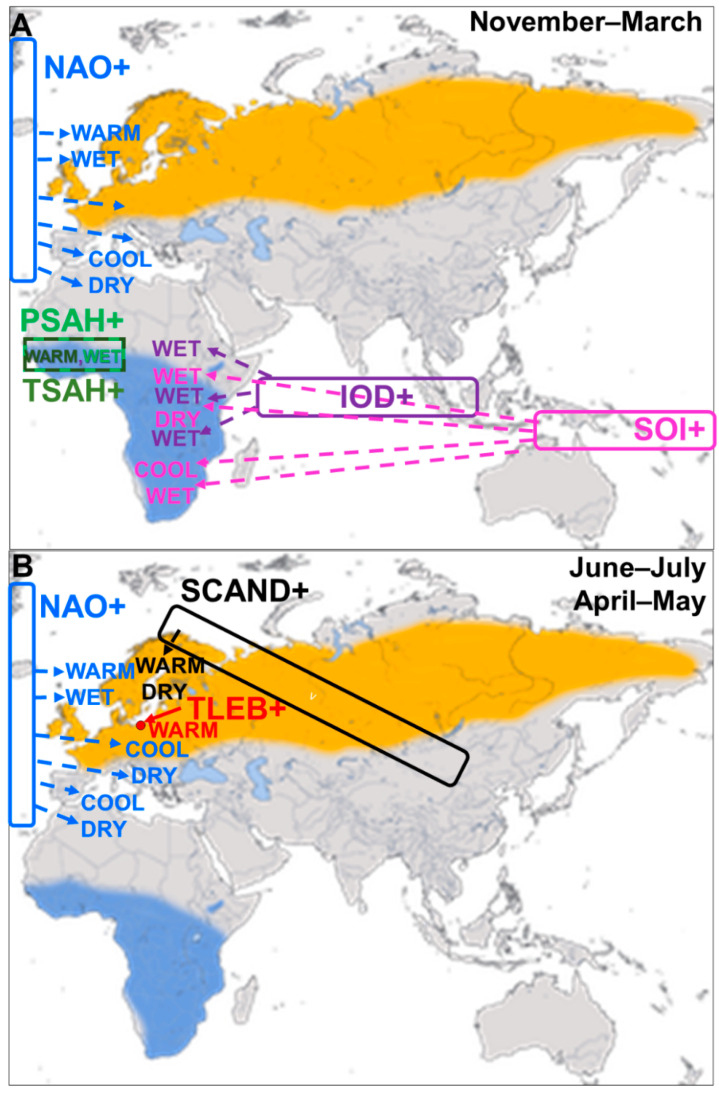
The approximate conditions at areas visited by the long-distance Euro-African migrants related with the positive phases of the analysed atmospheric patterns: (**A**) in November–March at both hemispheres, (**B**) in March–April and June–July at the northern hemisphere. Rectangles join the opposite centres of each climate pattern. Arrows and their labels indicate approximate areas influenced by each pattern and related conditions. Symbols of the climate indices as in Table 1. The climate indices are shown in relation to the geographical range of an example of a long-distance migrant, the Willow Warbler *Phylloscopus trochilus* (map by Andreas Trepte after [55], modified). The negative phases have opposite effects on the environment.

**Figure 5 animals-12-01732-f005:**
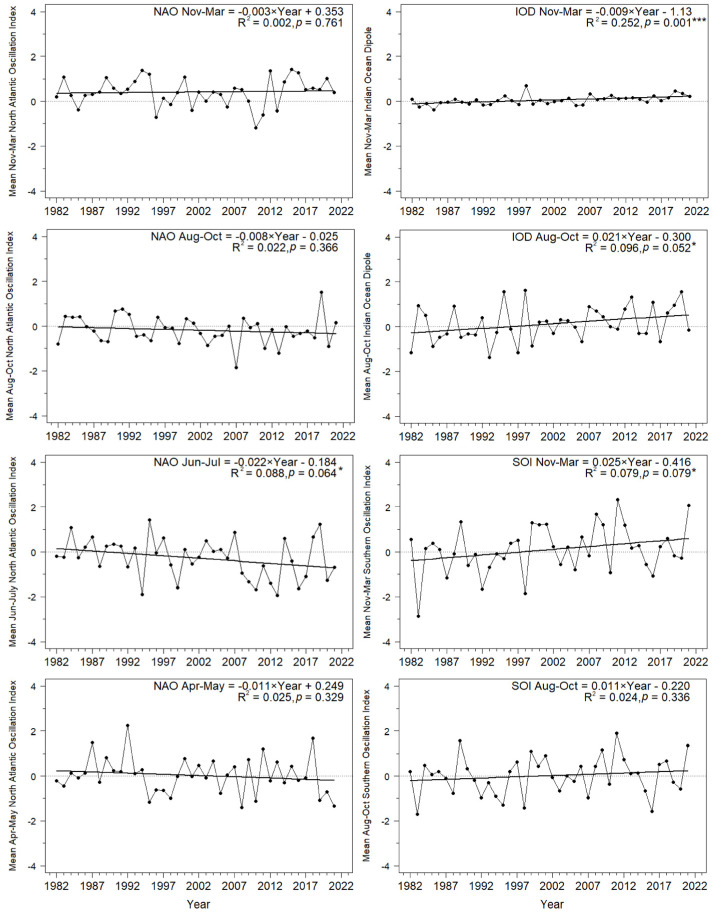
Trends and variation for the climate indices used in the study over 1981–2021. Year is used in the equations as the explanatory variable. Statistical significance of the regression equations: * 0.05 < *p* < 0.1, ** *p* < 0.01, *** *p* < 0.001.

**Figure 6 animals-12-01732-f006:**
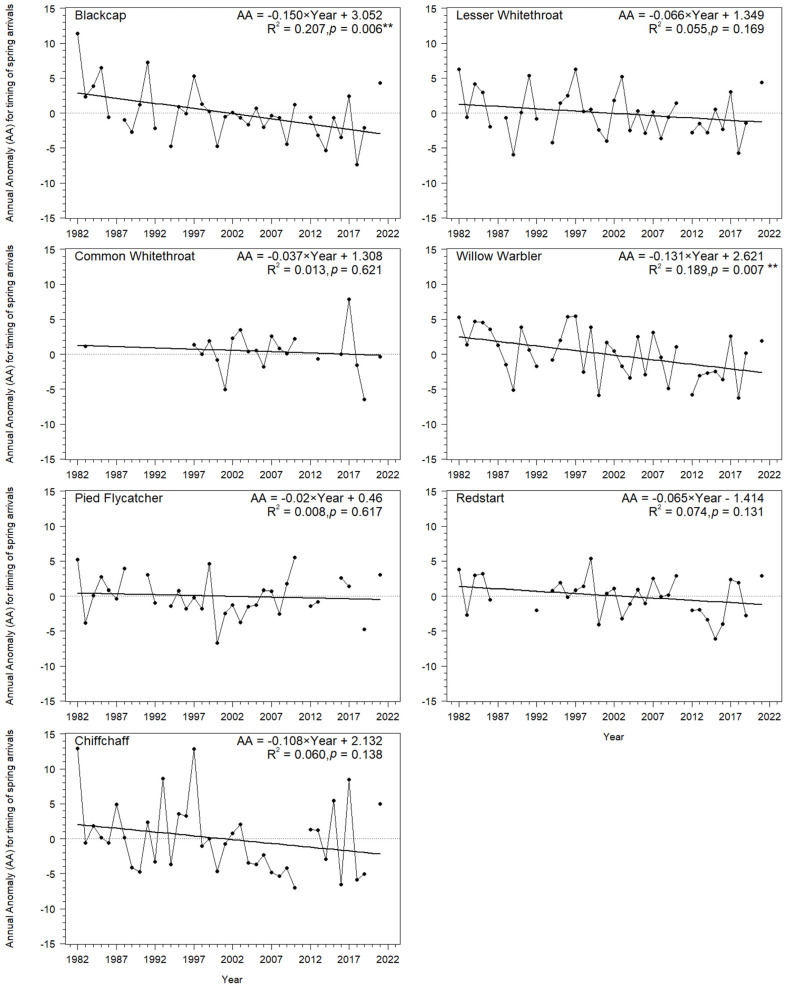
Trends for the Annual Anomaly (AA) of spring migration of the seven long-distance migrants at Bukowo, Poland, over 1982–2021. Year is used in the equations as the explanatory variable. Statistical significance of the regression equations: ** *p* < 0.01.

**Figure 7 animals-12-01732-f007:**
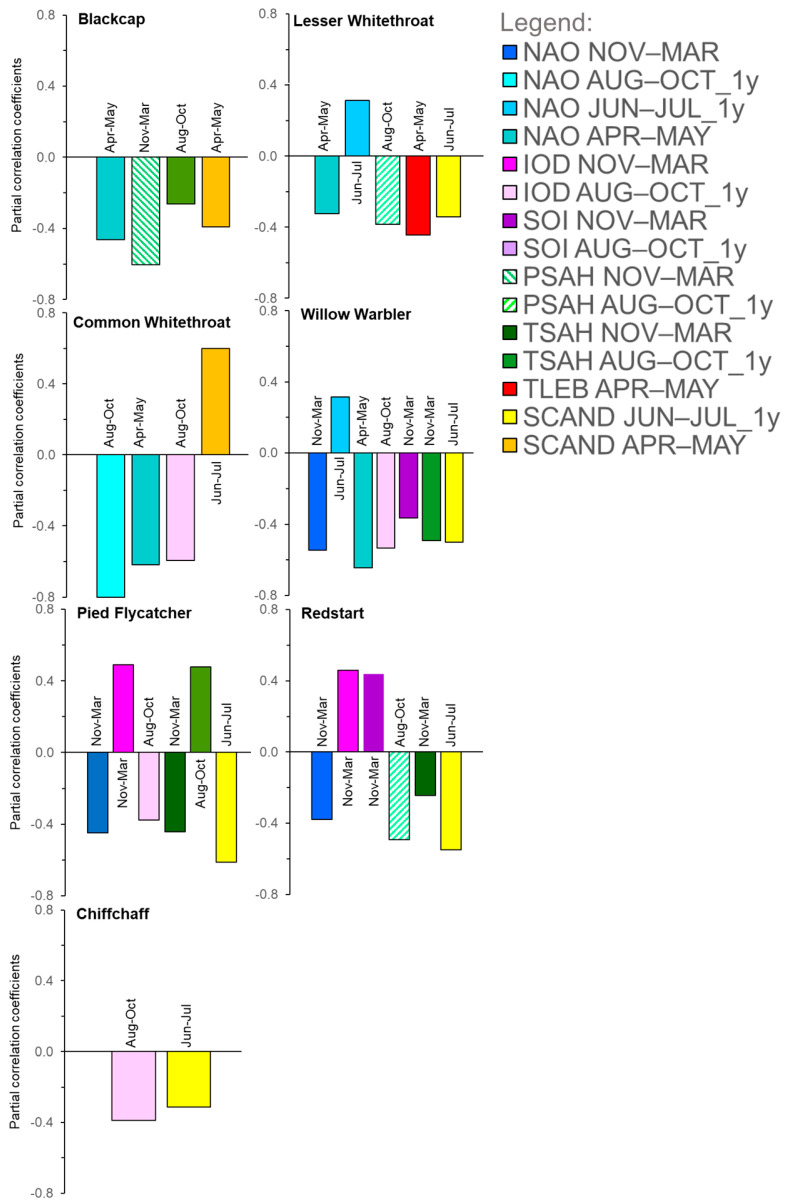
Partial correlation coefficients for Annual Anomaly (AA) of spring passage at Bukowo, Poland, over 1982–2021, against the climate indices selected in the best multiple regression models. The details of these models are presented in Appendix B, Table A12, Table A13, Table A14, Table A15, Table A16, Table A17 and Table A18.

**Table 1 animals-12-01732-t001:** Climate indices used as explanatory variables in modelling the timing of spring passage (23 March–15 May) of long-distance migrants over 1982–2021 at Bukowo, Poland. The local temperatures in Łeba (TLEB) were observations from the local weather station in Łeba; the remaining climate variables were reanalysed spatial data based on observations from weather stations and weather models, provided by the weather services at the listed sources.

Symbol ^1^	Climate Index	Months ^2^	Baseline Period ^3^	Source of Data
NAO NOV–MAR	North Atlantic Oscillation Index	Nov–Mar	1982–2021	https://www.cpc.ncep.noaa.gov/products/precip/CWlink/pna/norm.nao.monthly.b5001.current.ascii.table (accessed on 30 June 2022)
NAO AUG–OCT_1y		Aug–Oct_1y		
NAO JUN–JUL_1y		Jun–Jul_1y		
NAO APR–MAY		Apr–May		
IOD NOV–MAR	Indian Ocean Dipole	Nov–Mar	1981–2010	https://psl.noaa.gov/gcos_wgsp/Timeseries/DMI/ (accessed on 30 June 2022)
IOD AUG–OCT_1y		Aug–Oct_1y		
SOI NOV–MAR	Southern Oscillation Index	Nov–Mar	1981–2010	https://psl.noaa.gov/gcos_wgsp/Timeseries/SOI/ (accessed on 30 June 2022)
SOI AUG–OCT_1y		Aug–Oct_1y		
PSAH NOV–MAR	Sahel Precipitation Index within 10°–20° N, 20° W–10° E	Nov–Mar	1982–2021	https://climexp.knmi.nl/select.cgi?gpcc (accessed on 30 June 2022)
PSAH AUG–OCT_1y		Aug–Oct_1y		
TSAH NOV–MAR	Sahel temperature anomaly within 10°–20° N, 20° W–10° E	Nov–Mar	1982–2021	http://climexp.knmi.nl/select.cgi?era5_t2m_daily (accessed on 30 June 2022)
TSAH AUG–OCT_1y		Aug–Oct_1y		
TLEB APR–MAY	local temperatures in Łeba	Apr–May	1982–2021	http://www.ecad.eu (accessed on 30 June 2022)
SCAND JUN–JUL_1y	Scandinavian Pattern Index	Jun–Jul_1y	1981–2010	ftp://ftp.cpc.ncep.noaa.gov/wd52dg/data/indices/scand_index.tim (accessed on 30 June 2022)
SCAND APR–MAY		Apr–May		
Year		1982 = Year 1	1982–2021	Our database

^1^ Symbols _1y indicate the months of the year before the analysed spring migration. ^2^ Months for which the mean values of the listed climate indices were calculated. ^3^ Baseline periods differ because of the methods used to calculate each climate variable, as explained in each data source.

## Data Availability

Data supporting reported results can be found at the Global Biodiversity Information Facility database at: Ringing Data from the Bird Migration Research Station, University of Gdańsk (gbif.org) [145].

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
