# Peer review of "Large-Scale Climatic Patterns Have Stronger Carry-Over Effects than Local Temperatures on Spring Phenology of Long-Distance Passerine Migrants between Europe and Africa"

_animals, 2022, doi:10.3390/ani12131732_

Round 1

Reviewer 1 Report

All my comments are in the attached pdf-file, except an extract from the Handbuch der Vögel Mitteleuropas, which follows here:

Glutz von Blotzheim, Urs N. & Bauer, Kurt M. (1991), Aula, Wiesbaden

HANDBUCH DER VÖGEL MITTELEUROPAS BAND 12/II

Passeriformes (3. Teil): Sylviidae (Grasmücken, Laubsänger, Goldhähnchen)

Willow Warbler

Ankunft (p. 1328 bottom): NE-europäische Ph. t. acredula erreichen Jogeva/Estland am 21. April bis 15. Mai (mean of 34   4. Mai), Kemi-Tornie/Finnisch-Lappland am 3.–13. (mean of 13   9. Mai) und die Station Ladoga/Karelien (62°20 N/34° E) ab 7. Mai (TAMM, Orn. kogumik 5, 1971; RAUHALA 1.c.; LAPSCHIN 1978 zit. SCHÖNFELD 1982).

Reviergründung (p. 1336): Zwischen Eintreffen im Brutgebiet und Besetzung des späteren Territoriums vergehen meist 1–2, bei Eintritt längerer Schlechtwetterperioden mit Kälte und Schneeschauern auch bis zu 14 Tage (HOWARD 1907–191412*; LAWN, Brit. Birds 73, 1980 und 1982; SCHÖNFELD 1982). Die Ankunft aller dauert in Südengland etwa 4 (MAY 1949), in SE-Schottland 3 (BROCK, Zoologist  14, 1910) und in Südfinnland und Karelien 2-3 Wochen (KUUSISTO 1941; LAPSCHIN 1978 zit. SCHÖNFELD 1982), verläuft aber rascher, wenn der Zugbeginn durch schlechtes Wetter verzögert worden ist.

In einer während 7 Jahren kontrollierten mittelschwedischen Population dauerte der Revierbezug brutortstreuer 10–15 Tage, jener von neuen (wahrscheinlich einjährigen) > 3 Wochen; nach Ankunft von 90% der alten fehlten immer noch 50% der Erstsiedler (JAKOBSSON 1988). Die ersten erscheinen 1–3 Wochen nach den frühesten (HOWARD 1907–191412*, BROCK 1.c., MILDENBERGER 1940 u.a.).

Die Paarbildung erfolgt innerhalb von Stunden nach Ankunft des ; Balz setzt erst 2–3 Tage später ein (MAY 1949). Nach RADESÄTER u.a. (1987) und RADESÄTER & JAKOBSSON (1988 und 1989).

Nestbau: 5-7 Tage.

Brutperiode (p. 1336): Legebeginn gewöhnlich 2–3 Tage nach Vollendung des Nest-Mittelbaues (SCHÖNFELD 1982); in West- und Mitteleuropa Mitte April/Anfang Mai, Fennoskandien Ende

Mai/Mitte Juni.

In Finnland beginnt die Eiablage normalerweise in den letzten Mai- oder ersten Junitagen (69% von 139 Nestern zwischen 26. Mai und 9. Juni, das früheste vor dem 15. Mai), mit einer durchschnitt­lichen Verspätung von 1,0–1,5 Tagen je Breitengrad. Nachgelege werden noch bis Ende Juni, ausnahmsweise sogar bis Mitte Juli abgelegt (VON HAARTMAN 1.c.). In Lammi (61° N) Legebeginn in 121 Nestern zwischen 24. Mai und 11. Juli (Median um den 3./4. Juni), in Kemi-Tornio (65°45 N) zwischen 25. Mai und Anfang Juli (meist zwischen 5. und 15. Juni), in Värriö (66°50 N) zwischen 31. Mai und 6. Juli (Median87 14. Juni; TIAINEN 1983b, RAUHALA, Kemin-Tornion seudun linnusto, Kemi 1980, PULLIAINEN & SAARI Mskr.). Auf Süd-Jamal erfolgte die Eiablage in 59 Nestern zwischen 18. Juni und 8. Juli (Median für 43 vollendete Gelege um den 28.–30. Juni; DANILOW, RYSCHANOWSKIJ & RJABIZEW 198410*).

Brutdauer (p. 1341): 12–14(–15), durchschnittlich und unter normalen Bedingungen 13 Tage. – Nestlingsdauer 12–14(–15) Tage, wobei die Jungen bei Störungen das Nest auch schon im Alter von 10/11 Tagen verlassen können und mit 13 Tagen leidlich flugfähig sind (VON HAARTMAN 196910*, TIAINEN 1978 und 1983b, PULLIAINEN & SAARI Mskr.). Legeintervall 24 Stunden, Eiablage frühmorgens; Bebrütungsbeginn mit dem letzten, manchmal auch mit dem vorletzten Ei (MILDENBERGER 1940, MAY 1947).

Nach dem Flüggewerden (p. 1352) bleiben Fitis-Familien 2–3 Wochen zusammen (DA PRATO, Brit. Birds 74, 1981) und mit etwa 25 Tagen verlassen die Jungen das Elternrevier (NORMAN 1981b und 1985).

S. 1340 unten: – Sommerliche Verweildauer im Brutgebiet 20–22 Wochen (4 ½ to  5 months); this seems a bit too long for Scandinavia.

Simplified summary (for Finland): mean arrival ~ 8. May, territory establishment 1-3 weeks, females arrive 1-3 weeks later than males (i.e. after establishment of territories), pairing 3 d, nesting 5-7 days, laying 5 days, incubation 13 days, fledging after 13 days, fledglings leaving family group after 2-3 weeks. Juveniles starting migration 65-68 days after hatching.

Thus arrival ~8. May   + 14 d, +3, +6, +5, +13 (41 days from arrival until hatching of young)
+13 until fledging; fledglings leave the family after about 25 days, and leave the breeding area at an age of 65-68 days.
From hatching till migration of young 13+67 = 80.
Time from arrival until migration of young 41+80
à 121 days = 4 months.

4 months seem to be a reasonable assumption for breeding time in Scandinavia; 3 months might be a minimum; 2 months are impossible! 

Author Response

We are very grateful for thorough and helpful comments, which helped to improve the manuscript. To address the issue of the “oversimplified” life-cycle of a migrant. we rewrote the relevant passages in the text and reworked Figure 2 (now Figure 3) accordingly, following reviewer’s suggestions. We added required explanations on the statistics in the chapter 2.6. and added tables presenting the results from full models for each species (Appendix B, Tables B1-B7). We also added explanations that the analysis considers both the long-term trends and year-to-year variation in the places in the text where it was unclear. We formulated results and conclusions more cautiously, following all suggestions. We could not include Garden Warbler, Spotted Flycatcher and Red-backed Shrikes in our analyses as they are caught in spring at Bukowo in low numbers. We analysed all long-distance migrants for which we had sufficient sample sizes from spring passage at Bukowo. We rearranged all the figures and tables showing climate indices in the consistent order, matching the order in which they are discussed, as suggested. We rearranged Figure 1 with maps, and rearranged all figures and tables showing the species in a consistent order, matching that at the Figure 1.  We implemented all the suggested corrections in the text and in figures and tables. Detailed point-by-point answers to each comment are in the attached .pdf file. 

Reviewer 2 Report

The manuscript “Large-scale climatic patterns have combined effects on spring phenology of long-distance passerine migrants between Europe and Africa” is a manuscript of good quality with interesting analyses performed.

To this reviewer, the manuscript is acceptable for publication on ‘Animals’, however some minor revisions and clarifications are needed, as follows. 

General comments:

Review the English throughout the manuscript and simplify the sentence in some points (e.g. lines 92-95, line 184). Verify the style of references: for example in reference 55 remove “http” that is reported twice.

Please verify the correct citation of tables and figures throughout the manuscript.

Specific comments:

Figure 1: I suggest reorganising the maps in the figure, to put the maps with similar zoom close to each other. Consider to move the Willow Warbler’s map on the right, in the position now occupied by Redstart’s map. This is suggested to have the first six panels on the left of the same size and the same zoom (possibly), and the longer map alone over the legend. Consider to add a scale bar, as you mention distances in the manuscript.

Lines 147-149: unclear, please rephrase.

Lines 150-154: unclear, please add a graph showing the curves and better explain the way you average across the years.

Lines 178-180: unclear, please rephrase.

Figure 3: please describe somewhere in the text the parts A and B of the figure, and mention A and B in the caption of the figure. Please mention 3A or 3B across the manuscript, when referring to the figure. The content of the figure is not clear itself and how it is connected to lines 194-196 “We used a combination of these large-scale climatic indices, and local temperature and precipitation values for areas where large-scale indices were not available”. Moreover, it is not clear why only a species is mentioned in the caption: if it is an example, please report “e.g.”

Line 215: probably you mean Figure 3 instead of Figure 2?

Table 1. It is not clear to this reviewer which kind of data are the table 1 referring to. Are data referring to a spatial dataset (raster) or are they csv values for specific locations? Are they coming from remotely sensed imagery or from weather stations? Please add a sentence in the text or a column in the table to better explain the type of data used, as it is only clear in the acknowledgement!

Lines 237-239: please rephrase the sentence; it is too long and results confusing. Is it “cause” or “causes”?

Lines 253-254: the symbology mentioned in the caption is difficult to see in the graphs. Please improve readability.

Line 317: Is it correct Figure 3, mentioned here?

Line 304: please add “Appendix B”, before “Table B2”. Please complete in any other occurrence throughout the manuscript, when referring to the tables of the appendixes.

Line 399: please check the number of the figure and the tables.

Line 401: “two” species resulted statistically significant in their association with AA

Line 413: a full stop is missing at the end of the sentence

Line 779: table B2 in Appendix B is repeated twice.

Lines 749-753: the year 2011 is not mentioned. Please add a sentence explaining why (draught year with few bird catchments) 

Please verify that the references number 103, 104 and 105 are reported in the manuscript

Figure A1 in Appendix A, in the left map on the top, please consider to add the borders of the countries, or remove it, as the right map on the top is very similar. In the Google Earth image of the two locations, please consider to zoom in, as it is not clear where is the humid area and its extension. Please revise and improve the usefulness of the figure.

Figure A2 in Appendix A is visible only in the pdf file, not in the word file. Furthermore, it overlaps the figure A1.

Author Response

We are very grateful for thorough and helpful comments, which helped to improve the manuscript. The detailed comments are listed point by point in the attached file.

Reviewer 3 Report

Comments on Remisiewicz, M and Underhill, L.G. (submitted) Large-scale climatic patterns have combined effects on spring phenology of long-distance passerine migrants between Europe and Africa

General comments

This is quite an important paper which offers significant new insights on how large-scale climate influences phenology of long-distance Africa-Europe migrant birds. The paper draws on long-term data covering seven species over many decades. The analysis appears rigorous and the conclusions seem both sound and noteworthy. There were some slips in the original submitted manuscript, but these have been picked up and addressed in the updated version. The paper is quite clearly presented. The study is embedded well within the relevant literature with adequate introduction and discussion. Both authors have established track-records in this field. In this latest version, I have not detected any critical issues which would require substantial reworking, but I have offered various suggestions which I hope will be useful in improving the manuscript further.

Specific comments

The title doesn't really do the paper full justice. It would be good to have a title which refers to the key finding that spring phenology of long-distance passerine migrants can be influenced not just by local weather conditions upon their spring arrival in Europe, but by prior weather conditions throughout their annual cycles over distant wintering grounds and migration routes.    

Page 1, line 32 '..The best models..'

Page 4, line 161 '..migration takes four..'

From the analytical methods outlined in Section 2.6, together with the explanations of climate indices given in the subsections of Section 2.5, I think the analytical implications of trends in both phenology and the indices have been adequately taken into consideration. I do think it might be useful though to give the reader just a little more explanation on this.

There were some issues with Figure 5 in the original version, but these have now been addressed and the results of the paper all fit together and make more sense now. 

It might be good to do just a couple of further very minor edits to Figure 5 and its legend. I think the Annual Anomalies refer to spring arrival dates rather than spring itself. If there is space to somehow rework the axis labels in that regard then I think that might be better. In the legend, spell out '..long-distance..'. If Figure 5 only refers to arrival dates and does not refer to climate indices, then the period of the climate indices does not need to be mentioned in the legend. If it is mentioned, there are two minor typos: '..The climate...cover 1981-..'. The explanation of how statistical significance is denoted is probably not needed in the legend, particularly when two of the three significance categories are not actually used. On Figure 5 itself, if it is easy to shift the '**' up next to the p-value in the top right of each box, that might be clearer. In its current position on the graph itself, it can at first easily be mistaken for two extra data points. 

The results for redstarts in the first version were at odds with the other species and with findings from other studies but this revised version has checked and tackled some issues there and the findings are now more consistent.

(In the revised version, as well as edits to the text itself, there are also some comments which state what changes have been made. That's fine of course, but I do hope it is clear which pieces of text will be published and which pieces are only there as explanation for those involved in handling the manuscript. In the revised MS-Word version, the changes and comments are flagged in red, but in the pdf version the comments are in black and indistinguishable from regular text so there is a risk that the published version may not be clean and tidy.)

Page 12 (now 14 in the revised version). Would it be worth doing some kind of test across the multiple species? I'm not sure a formal meta-analysis is needed, but some kind of multi-species test might be informative given that all seven slopes have negative sign, even if only two are individually significant when analysed separately.

Page 13 (now 15), discussion. I'm not sure there is necessarily a discrepancy between this study and other studies which have found significant advances in phenology for the remaining species. The trends may be non-significant here but that needn't mean they differ significantly from other studies which did detect significant trends.   

Page 15 (now 17), Section 3.2.2. I may have got mixed up between the different shades of blue in Figure 6, but I'm not sure I can match up the things I see in Figure 6 with the interpretations about NAO given in Section 3.2.2.(?) 

(In the corrected version, some of the original page-breaks, paragraph spacing, etc, have been shunted so a little re-tidying will be needed. For example, much of Page 13 is now blank, and e.g. there is no spacing between Section 3.2.2, 3.2.3, 3.2.4 etc.)

Page 17, Section 3.2.3, line 455. I suggest rephrasing a little and saying that "..spring passage was related with IOD in August-October and/or November to March..". If I have interpreted Figure 6 correctly then redstarts are affected significantly by IOD only in Nov-Mar, chiffchaffs, whitethroats and willow warblers are significantly affected by IOD only in Aug-Oct, and pied flycatchers are significantly affect by IOD in both Aug-Oct and Nov-Mar.

Pages 19 and 20, Section 3.2.6. If I have understood correctly, the data in this paper do refer to 'arrival' dates of migrants in Europe (more specifically northern Poland), but this is not necessarily arrival on the breeding grounds per se. It might be worth just inserting some brief mention/discussion of this. I imagine migrants caught at Bukowo may be drawn from fairly wide breeding areas(?), though I also imagine the exact breeding areas might not be precisely mapped yet(?). Local weather conditions at Bukowo may have little influence on the dates of arrival in Bukowo but I guess they could have some subsequent knock-on influence on when they reach the breeding grounds per se(?), even if these are not being examined here. (If so, then I guess it is also not inconceivable that Bukowo weather in year t-1 could affect arrival dates at Bukowo in year t?)

Page 21, Section 3.4, lines 667-669. I suggest rephrasing this sentence to make it clearer.

The figures and tables at the end of the paper are certainly detailed and extensive so readers will be able to examine many points in depth.

Author Response

(The authors gave the same response as above.)
